# SSL-Lanes: Self-Supervised Learning for Motion Forecasting in Autonomous Driving

**Prarthana Bhattacharyya**    **Chengjie Huang**    **Krzysztof Czarnecki**

University of Waterloo, Canada
`{p6bhatta, c.huang, k2czarne}@uwaterloo.ca`

**Abstract:** Self-supervised learning (SSL) is an emerging technique that has been successfully employed to train convolutional neural networks (CNNs) and graph neural networks (GNNs) for more transferable, generalizable, and robust representation learning. However its potential in motion forecasting for autonomous driving has rarely been explored. In this study, we report the first systematic exploration and assessment of incorporating self-supervision into motion forecasting. We first propose to investigate four novel self-supervised learning tasks for motion forecasting with theoretical rationale and quantitative and qualitative comparisons on the challenging large-scale Argoverse dataset. Secondly, we point out that our auxiliary SSL-based learning setup not only outperforms forecasting methods which use transformers, complicated fusion mechanisms and sophisticated online dense goal candidate optimization algorithms in terms of performance accuracy, but also has low inference time and architectural complexity. Lastly, we conduct several experiments to understand why SSL improves motion forecasting.

**Keywords:** Motion Forecasting, Autonomous Driving, Self-Supervised Learning

## 1 Introduction

Motion forecasting in a real-world urban environment is an important task for autonomous robots. It involves predicting the future trajectories of traffic agents including vehicles and pedestrians. This is absolutely crucial in the self-driving domain for safe, comfortable and efficient operation. However, this is a very challenging problem. Difficulties include inherent stochasticity and multimodality of driving behaviors, and that future motion can involve complicated maneuvers such as yielding, nudging, lane-changing, turning and acceleration or deceleration.

The motion prediction task has traditionally been based on kinematic constraints and road map information with handcrafted rules. These approaches however fail to capture long-term behavior and interactions with map structure and other traffic agents in complex scenarios. Tremendous progress has been made with data-driven methods in motion forecasting [3, 4, 5, 6, 7, 8, 9, 10]. Recent methods use a vector representation for HD maps and agent trajectories, including approaches like Lane-GCN [2], Lane-RCNN [11], Vector-Net [12], TNT [5] and Dense-TNT [6]. More recently, the enormous success of transformers [13] has been leveraged for forecasting in mm-Transformer [9], Scene transformer [8], Multimodal transformer [14] and Latent Variable Sequential Transformers [15]. Most of these methods however are extremely complex in terms of architecture and have low inference speeds, which makes them unsuitable for real-world settings.

In this work, we extend ideas from self-supervised learning (SSL) to the motion forecasting task. Self-supervision has seen huge interest in both natural language processing and computer vision [16] to make use of freely available data without the need for annotations. It aims to assist the model to learn more transferable and generalized representation from pseudo-labels via pretext tasks. Given the recent success of self-supervision with CNNs, transformers, and GNNs, we are naturally motivated to ask the question: *Can self-supervised learning improve accuracy and generalizability of motion forecasting, without sacrificing inference speed or architectural simplicity?*

**Contributions:** Our work, SSL-Lanes, presents the first systematic study on how to incorporate self-supervision in a standard data-driven motion forecasting model. Our contributions are: (a)

6th Conference on Robot Learning (CoRL 2022), Auckland, New Zealand.

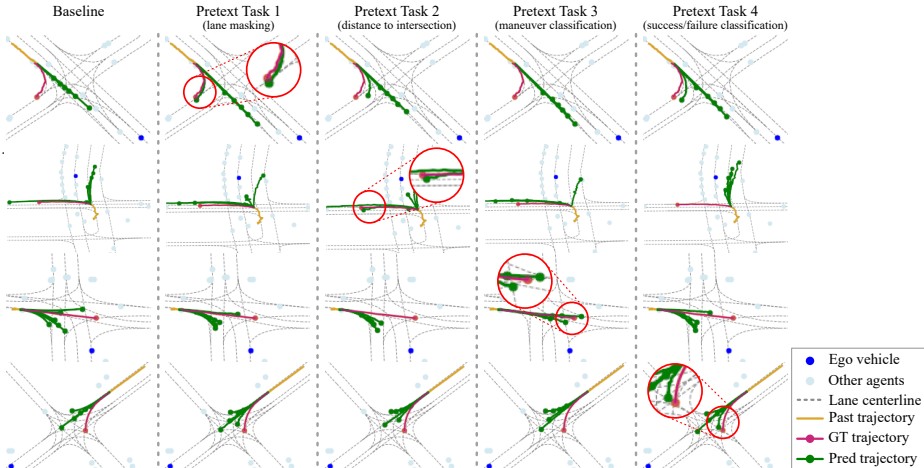

Figure 1: Motion forecasting on Argoverse [1] validation. We show four challenging scenarios at intersections. The baseline [2] misses all the predictions. In the first row, our proposed lane masking successfully captures the right-turn. For the second row, predicting distance to intersection helps the most in capturing the left turn. In the third row, acceleration at an intersection is best captured by the model that is made to classify maneuvers of traffic agents. Finally, in the fourth row, classifying successful final goal states is the most effective at capturing the left turn. These tasks are trained with pseudo-labels which are obtained for free from data.

We demonstrate the effectiveness of incorporating self-supervised learning in motion forecasting. Since this does not add extra parameters or compute during inference, SSL-Lanes achieves the best accuracy-simplicity-efficiency trade-off on the challenging large-scale Argoverse [1] benchmark. (b) We propose four self-supervised tasks based on the nature of the motion forecasting problem. The key idea is to leverage easily accessible map/agent-level information to define domain-specific pretext tasks that encourage the standard model to capture more superior and generalizable representations for forecasting in comparison to pure supervised learning. (c) We further design experiments to explore why forecasting benefits from SSL. We provide extensive results to hypothesize that SSL-Lanes learns richer features from the SSL training as compared to a model trained with vanilla supervised learning.

## 2   Related Work

**Motion Forecasting:** Traditional methods for motion forecasting primarily use Kalman filtering [17] with a prior from HD-maps to predict future motion states [18, 19]. With the huge success of deep learning, recent works use data-driven approaches for motion forecasting. These methods explore different architectures involving rasterized images and CNNs [3, 20, 21], vectorized representations and GNNs [12, 11, 22, 4, 7], point-cloud representations [23], transformers [8, 9, 15, 14] and sophisticated fusion mechanisms [2], to generate features that predict final output trajectories. While the focus of these works is to find more effective ways of feature extraction from HD-maps and interacting agents, they need huge model capacity, heavy parameterization, and extensive augmentations or large amounts of data to converge to a general solution. Other works [5, 10, 24, 25] build on them to incorporate prior knowledge in the form of predefined candidate trajectories obtained from sampling or clustering strategies from training data. However the disadvantage of these methods is that their performance is highly related to the quality of the trajectory proposals, which becomes an extra dependency. End-to-end solutions for optimizing end-points of these candidates trajectories are proposed by Dense-TNT [6] and HOME [26]. Dense-TNT has state-of-the-art accuracy with a reasonable parameter budget, but its online dense goal candidate optimization strategy is computationally very expensive, which is unrealistic for real-time operations like autonomous driving. Lately, ensembling techniques like MultiPath++ [27] and DCMS [28] have been proposed and while they have high forecasting performance, a major disadvantage is their high memory cost for training and heavy computational cost at inference.

**Self-supervised Learning:** SSL is a rapidly emerging learning framework that generates additional supervised signals to train deep learning models through carefully designed pretext tasks. In the

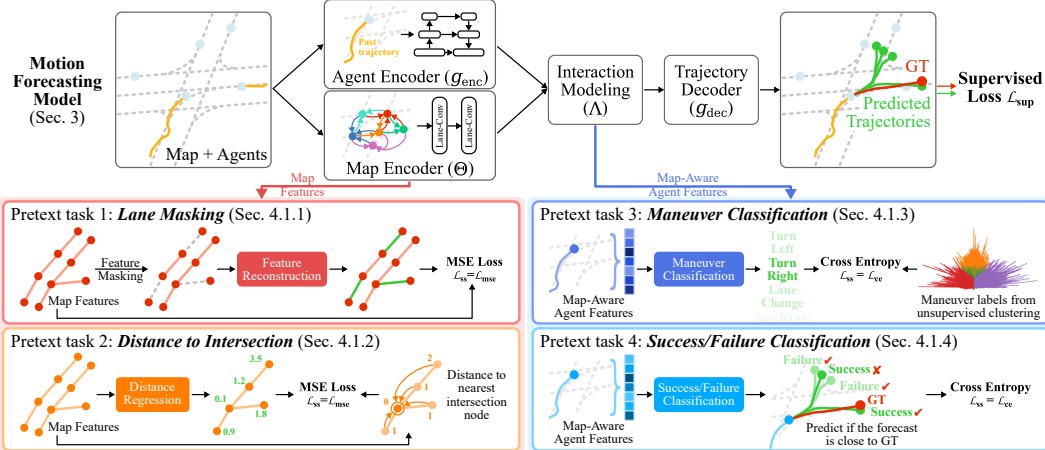

Figure 2: Illustration of the overall SSL-Lanes framework for self-supervision on motion forecasting through joint training. SSL-Lanes improves upon a standard-motion forecasting baseline, that consists of an agent encoder, map encoder, interaction model and a trajectory decoder, trained using a supervised loss $\mathcal{L}_{sup}$. SSL-Lanes proposes four pretext tasks: (1) Lane Masking: which recovers feature information from the perturbed lane graphs. (2) Distance to Intersection: which predicts the distance (in terms of shortest path length) from all lane nodes to intersection nodes. (3) Maneuver Classification: predicts the form of a 'maneuver' the agent-of-interest intends to execute (4) Success/Failure Classification: which trains an agent specialized at achieving end-point goals.

image domain, various self-supervised learning techniques have been developed for learning high-level image representations, including predicting the relative locations of image patches [29], jigsaw puzzle [30], image rotation [31], image clustering [32], image inpainting [33], image colorization [34] and segmentation prediction [35]. In the domain of graphs and graph neural networks, pretext tasks include graph partitioning, node clustering, context prediction and graph completion [36, 37, 38, 39]. To the best of our knowledge, this is the first principled approach that explores motion forecasting for autonomous driving with self-supervision.

## 3 Background

**Problem Formulation:** We are given the past motion of $N$ actors. The $i$-th actor is denoted as a set of center locations over the past $L$ time-steps. We pre-process it to represent each trajectory as a sequence of displacements $\mathcal{P}_i = \{\Delta \boldsymbol{p}_i^{-L+1}, ..., \Delta \boldsymbol{p}_i^{-1}, \Delta \boldsymbol{p}_i^0\}$, where $\boldsymbol{p}_i^l$ is the 2D displacement from time step $l-1$ to $l$. We are also given a high-definition (HD) map, which contains lanes and semantic attributes. Each lane is composed of many consecutive lane nodes, with a total of $M$ nodes. $\boldsymbol{X} \in \mathbb{R}^{M \times F}$ denotes the lane node feature matrix, where $\boldsymbol{x}_j = \boldsymbol{X}[j, :]^T$ is the $F$-dimensional lane node vector. Following the connections between lane centerlines (i.e., predecessor, successor, left neighbour and right neighbour), we represent the connectivity within the lane nodes with 4 adjacency matrices $\{\boldsymbol{A}_f\}_{f \in \{\text{pre,suc,left,right}\}}$, with $\boldsymbol{A}_f \in \mathbb{R}^{M \times M}$. This implies that if $\boldsymbol{A}_{f,gh} = 1$, then node $h$ is an $f$-type neighbor of node $g$. Our goal is to forecast the future motions of all actors in the scene $\mathcal{O}_{\text{GT}}^{1:T} = \{(x_i^1, y_i^1), ..., (x_i^T, y_i^T) | i = 1, ..., N\}$, where $T$ is our prediction horizon.

**Standard Motion Forecasting Model:** We briefly introduce a standard data-driven motion forecasting framework, consisting of a feature encoder, interaction-modeler and prediction header.

*Feature Encoding:* We first encode the agent and map inputs similar to Lane-GCN [2]. The agent encoder includes a 1D convolution with a feature pyramid network, parameterized by $g_{\text{enc}}$, as given by Eq. (1). For map-encoding, we adopt two Lane-Conv residual blocks, parameterized by $\boldsymbol{\Theta} = \{\boldsymbol{W}_0, \boldsymbol{W}_{\text{left}}, \boldsymbol{W}_{\text{right}}, \boldsymbol{W}_{\text{pre},k}, \boldsymbol{W}_{\text{suc},k}\}$, where $k \in \{1, 2, 4, 8, 16, 32\}$, as given by Eq. (2).

$$\hat{\boldsymbol{p}}_i = g_{\text{enc}}(\mathcal{P}_i) \tag{1}$$

$$\boldsymbol{Y} = \boldsymbol{X}\boldsymbol{W}_0 + \sum_{j \in \{\text{left,right}\}} \boldsymbol{A}_j \boldsymbol{X}\boldsymbol{W}_j + \sum_k \boldsymbol{A}_{\text{pre}}^k \boldsymbol{X}\boldsymbol{W}_{\text{pre},k} + \boldsymbol{A}_{\text{suc}}^k \boldsymbol{X}\boldsymbol{W}_{\text{suc},k} \tag{2}$$

*Modeling Interactions:* Since the behavior of agents depends on map topology and social consistency, each encoded agent $i$ subsequently aggregates context from the surrounding map features and

| SSL Task | Property Level | Primary Assumption | Type |
|---|---|---|---|
| Lane-Masking | Map features | Local map structure | Aux. auto-encoder |
| Distance to Intersection | | Global map structure | Aux. regression |
| Maneuver Classification | Map-aware | Agent feature similarity | Aux. classification |
| Success/Failure Classification | agent features | Distance to success state | |

Table 1: Overview of our proposed self-supervised (SSL) tasks

its neighboring agent features, via spatial attention [40] as given by Eq. (3):

$$\tilde{\boldsymbol{p}}_i = \hat{\boldsymbol{p}}_i \boldsymbol{W}_{\text{M2A}} + \sum_j \phi(\text{concat}(\hat{\boldsymbol{p}}_i, \Delta_{i,j}, \boldsymbol{y}_j) \boldsymbol{W}_1) \boldsymbol{W}_2$$

$$\acute{\boldsymbol{p}}_i = \tilde{\boldsymbol{p}}_i \boldsymbol{W}_{\text{A2A}} + \sum_j \phi(\text{concat}(\tilde{\boldsymbol{p}}_i, \Delta_{i,j}, \tilde{\boldsymbol{p}}_j) \boldsymbol{W}_3) \boldsymbol{W}_4$$

(3)

Here, $\boldsymbol{y}_j$ is the feature of the $j$-th node, $\hat{\boldsymbol{p}}_i$ is the feature of the $i$-th agent, $\phi$ the composition of layer normalization and ReLU, and $\Delta_{ij} = \text{MLP}(\boldsymbol{v}_j - \boldsymbol{v}_i)$, where $\boldsymbol{v}$ denotes the $(x, y)$ 2-D bird's-eye-view (BEV) location of the agent or the lane node. The parameters for map and agent feature aggregation is represented by $\boldsymbol{\Lambda} = \{\boldsymbol{W}_{\text{M2A}}, \boldsymbol{W}_1, \boldsymbol{W}_2, \boldsymbol{W}_{\text{A2A}}, \boldsymbol{W}_3, \boldsymbol{W}_4\}$.

*Trajectory Prediction:* Finally, we decode the future trajectories from the features $\acute{\boldsymbol{p}}_i$ corresponding to the agents of interest as given by: $\mathcal{O}_{\text{pred}}^{1:T} = \{g_{\text{dec}}(\acute{\boldsymbol{p}}_i)|i = 1, ..., N\}$, where $g_{\text{dec}}$ is the parameterized trajectory decoder. The parameters for the motion forecasting model are learned by minimizing the supervised loss ($\mathcal{L}_{\text{sup}}$) calculated between the predicted output and the ground-truth future trajectories ($\mathcal{O}_{\text{GT}}^{1:T}$), as given by Eq. (4):

$$g_{\text{enc}}^{\star}, \boldsymbol{\Theta}^{\star}, \boldsymbol{\Lambda}^{\star}, g_{\text{dec}}^{\star} = \underset{g_{\text{enc}}, \boldsymbol{\Theta}, \boldsymbol{\Lambda}, g_{\text{dec}}}{\arg\min} \mathcal{L}_{\text{sup}}(\mathcal{O}_{\text{pred}}^{1:T}, \mathcal{O}_{\text{GT}}^{1:T})$$

(4)

## 4  SSL-Lanes

The goal of our proposed SSL-Lanes framework is to improve the performance of the primary motion forecasting baseline by learning simultaneously with various self-supervised tasks. Fig. 2 shows the pipeline of our proposed approach, and Tab. 1 summarizes the self-supervised tasks.

**Self-Supervision meets Motion Forecasting:** Considering our motion forecasting task and a self-supervised task, the output and the training process can be formulated as:

$$\boldsymbol{\Psi}^{\star}, \boldsymbol{\Omega}^{\star}, \boldsymbol{\Theta}_{\text{ss}}^{\star} = \underset{\boldsymbol{\Psi}, \boldsymbol{\Omega}, \boldsymbol{\Theta}_{\text{ss}}}{\arg\min} \quad \alpha_1 \mathcal{L}_{\text{sup}}(\boldsymbol{\Psi}, \boldsymbol{\Omega}) + \alpha_2 \mathcal{L}_{\text{ss}}(\boldsymbol{\Psi}, \boldsymbol{\Theta}_{\text{ss}})$$

(5)

where, $\mathcal{L}_{\text{ss}}(\cdot, \cdot)$ is the loss function of the self-supervised task, $\boldsymbol{\Theta}_{\text{ss}}$ parameterizes the corresponding task-specific layers, and $\alpha_1, \alpha_2 \in \text{R}_{>0}$ are the weights for the supervised and self-supervised losses. If the pretext task only focuses on the map encoder, then $\boldsymbol{\Psi} = \{\boldsymbol{\Theta}\}$ and $\boldsymbol{\Omega} = \{g_{\text{enc}}, \boldsymbol{\Lambda}, g_{\text{dec}}\}$. Otherwise, $\boldsymbol{\Psi} = \{g_{\text{enc}}, \boldsymbol{\Theta}, \boldsymbol{\Lambda}\}$ and $\boldsymbol{\Omega} = \{g_{\text{dec}}\}$. Henceforth, we also define the following representations. We will represent the primary task encoder as function $f_{\boldsymbol{\Psi}}$, parameterized by $\boldsymbol{\Psi}$. Furthermore, given a pretext task, which we will design in the next section, the pretext decoder $p_{\boldsymbol{\Theta}_{\text{ss}}}$ is a function that predicts pseudo-labels and is parameterized by $\boldsymbol{\Theta}_{\text{ss}}$.

### 4.1  Pretext tasks for Motion Forecasting

At the core of our SSL-Lanes approach is defining pretext tasks based upon self-supervised information from the underlying map structure *and* the overall temporal prediction problem itself (Tab. 1).

#### 4.1.1  Lane-Masking

The goal of the *Lane-Masking* pretext task is to encourage the map encoder $\boldsymbol{\Psi} = \{\boldsymbol{\Theta}\}$ to learn local structure information in addition to the forecasting task that is being optimized. Specifically, we randomly mask (i.e., set equal to zero) the features of $m_a$ percent of nodes per lane and then ask the self-supervised decoder to reconstruct these features.

$$\boldsymbol{\Psi}^{\star}, \boldsymbol{\Theta}_{\text{ss}}^{\star} = \underset{\boldsymbol{\Psi}, \boldsymbol{\Theta}_{\text{ss}}}{\arg\min} \frac{1}{m_a} \sum_{i=1}^{m_a} \mathcal{L}_{\text{mse}}\left(p_{\boldsymbol{\Theta}_{\text{ss}}}([f_{\boldsymbol{\Psi}}(\tilde{\boldsymbol{X}}, \boldsymbol{A}_f)]_{\boldsymbol{v}_i}), \boldsymbol{X}_i\right)$$

(6)

| Method | minADE$_1$ | minFDE$_1$ | MR$_1$ | minADE$_6$ | minFDE$_6$ | MR$_6$ |
|---|---|---|---|---|---|---|
| Baseline | 1.42 | 3.18 | 51.35 | 0.73 | 1.12 | 11.07 |
| Lane-Masking | 1.36 | 2.96 | 49.45 | **0.70** | 1.02 | 8.82 |
| Distance to Intersection | 1.38 | 3.02 | 49.53 | 0.71 | 1.04 | 8.93 |
| Maneuver Classification | **1.33** | **2.90** | 49.26 | 0.72 | 1.05 | 9.36 |
| Success/Failure Classification | 1.35 | 2.93 | **48.54** | **0.70** | **1.01** | **8.59** |

Table 2: Motion forecasting performance on Argoverse validation with our proposed pretext tasks

Here, $\tilde{X}$ is the node feature matrix corrupted with random masking, i.e., some rows of $X$ corresponding to nodes $v_i$ are set to zero. $p_{\Theta_{ss}}$ is a fully connected network that maps the representations to the reconstructed features. $\mathcal{L}_{mse}$ is the mean squared error (MSE) loss function penalizing the distance between the reconstructed map features $p_{\Theta_{ss}}([f_{\Psi}(\tilde{X}, A_f)]_{v_i})$ for node $v_i$ and its actual features $X_i$.

### 4.1.2 Distance to Intersection

*Distance-to-Intersection* pretext task is proposed to guide the map-encoder, $\Psi = \{\Theta\}$, to maintain global topology information by predicting the distance (in terms of shortest path length) from all lane nodes to intersection nodes. We aim to regress the distances from each lane node to pre-labeled intersection nodes annotated as part of the dataset. Given $K$ labeled intersection nodes $\mathcal{V}_{intersection} = \{v_{intersection,k} | k = 1, ...K\}$, we first generate reliable pseudo labels using breadth-first search (BFS). Specifically, BFS calculates the shortest distance $d_i \in \mathbb{R}$ for every lane node $v_i$ from the given set $\mathcal{V}_{intersection}$. The target of this task is to predict the pseudo-labeled distances using a pretext decoder. If $p_{\Theta_{ss}}([f_{\Psi}(X, A_f)]_{v_i})$ is the prediction of node $v_i$, and $\mathcal{L}_{mse}$ is the mean-squared error loss function for regression, then the loss formulation for this SSL pretext task is as follows:

$$\Psi^{\star}, \Theta_{ss}^{\star} = \underset{\Psi, \Theta_{ss}}{\arg\min} \frac{1}{M} \sum_{i=1}^{M} \mathcal{L}_{mse}\Big(p_{\Theta_{ss}}([f_{\Psi}(X, A_f)]_{v_i}), d_i\Big) \tag{7}$$

### 4.1.3 Maneuver Classification

We propose *Maneuver Classification*, and we expect it to provide prior regularization to $\Psi = \{g_{enc}, \Theta, \Lambda\}$, based on driving modes of agents. We aim to construct pseudo label to divide agents into different clusters according to their driving behavior and thus explore unsupervised clustering algorithms to acquire the maneuver for each agent. We find that using naive $k$-Means (on agent end-points) or DBSCAN (on Hausdorff distance between entire trajectories [41]) leads to noisy clustering. We find that constrained $k$-means [42] on agent end-points works best to divide trajectory samples into $C$ clusters equally. We define $C = \{maintain\text{-}speed, accelerate, decelerate, turn\text{-}left, turn\text{-}right, lane\text{-}change\}$ and the clustering function as $\rho$. If $p_{\Theta_{ss}}(f_{\Psi}(\mathcal{P}_i, X, A_f))$ is the prediction of agent $i$'s intention and $E_i = (x_{i,GT}^T, y_{i,GT}^T)$ is its ground-truth end-point, then the learning objective is to classify each agent maneuver into its corresponding cluster using cross-entropy loss $\mathcal{L}_{ce}$ as:

$$\Psi^{\star}, \Theta_{ss}^{\star} = \underset{\Psi, \Theta_{ss}}{\arg\min} \mathcal{L}_{ce}\Big(p_{\Theta_{ss}}(f_{\Psi}(\mathcal{P}_i, X, A_f)), \rho(E_i)\Big) \tag{8}$$

### 4.1.4 Forecasting Success/Failure Classification

We propose a pretext task called *Success/Failure Classification*, which trains an agent specialized at achieving end-point goals and thus links directly to the forecasting task. We expect this to constrain $\Psi = \{g_{enc}, \Theta, \Lambda\}$ to predict trajectories $\epsilon$ distance away from the correct final end-point. Similar to maneuver classification, we wish to create pseudo-labels for our data samples. We label trajectory predictions as successful ($c = 1$) if the final prediction $(x_{i,pred}^T, y_{i,pred}^T)$ is within $\epsilon < 2\,m$ of the final end-point $E_i$, and as failure ($c = 0$) otherwise. We choose $2\,m$ as our $\epsilon$ threshold because it is also used for miss-rate calculation (Sec. 5). If the pretext decoder predicts agent $i$'s final-endpoint as $p_{\Theta_{ss}}(f_{\Psi}(\mathcal{P}_i, X, A_f))$ and, given the ground-truth end-point $E_i$, its success or failure label is $c_i$,

| Method | minADE$_1$ | minFDE$_1$ | MR$_1$ | minADE$_6$ | minFDE$_6$ | MR$_6$ | b-FDE$_6$ |
|---|---|---|---|---|---|---|---|
| NN + Map [1] | 3.65 | 8.12 | 94.0 | 2.08 | 4.02 | 58.0 | - |
| Jean [4] | 1.74 | 4.24 | 68.56 | 0.98 | 1.42 | 13.08 | 2.12 |
| Lane-GCN [2] | 1.71 | 3.78 | 58.77 | 0.87 | 1.36 | 16.20 | 2.05 |
| LaneRCNN [11] | 1.68 | 3.69 | 56.85 | 0.90 | 1.45 | 12.32 | 2.15 |
| TNT [5] | 1.77 | 3.91 | 59.70 | 0.94 | 1.54 | 13.30 | 2.14 |
| DenseTNT [6] | 1.68 | 3.63 | 58.43 | 0.88 | 1.28 | 12.58 | 1.97 |
| PRIME [24] | 1.91 | 3.82 | 58.67 | 1.22 | 1.55 | 11.50 | 2.09 |
| WIMP [7] | 1.82 | 4.03 | 62.88 | 0.90 | 1.42 | 16.69 | 2.11 |
| TPCN [23] | 1.66 | 3.69 | 58.80 | 0.87 | 1.38 | 15.80 | 1.92 |
| HOME [26] | 1.70 | 3.68 | 57.23 | 0.89 | 1.29 | **8.46** | **1.86** |
| mmTransformer [9] | 1.77 | 4.00 | 61.78 | 0.87 | 1.34 | 15.40 | 2.03 |
| MultiModalTransformer [14] | 1.74 | 3.90 | 60.23 | 0.84 | 1.29 | 14.29 | 1.94 |
| LatentVariableTransformer [15] | - | - | - | 0.89 | 1.41 | 16.00 | - |
| SceneTransformer [8] | 1.81 | 4.06 | 59.21 | **0.80** | **1.23** | 12.55 | 1.88 |
| Success/Failure Classification (Ours) | **1.63** | **3.56** | **56.71** | 0.84 | 1.25 | 13.26 | 1.94 |

Table 3: Comparison of our (best) proposed model and top approaches on the Argoverse Test. The best results are in bold and underlined, and the second best is also underlined.

then the pretext loss can be formulated as:

$$\boldsymbol{\Psi}^\star, \boldsymbol{\Theta}_{ss}^\star = \underset{\boldsymbol{\Psi}, \boldsymbol{\Theta}_{ss}}{\arg\min} \, \mathcal{L}_{ce}\Big( p_{\boldsymbol{\Theta}_{ss}}(f_{\boldsymbol{\Psi}}(\mathcal{P}_i, \boldsymbol{X}, \boldsymbol{A}_f)), c_i \Big) \tag{9}$$

## 4.2 Learning

As all the modules are differentiable, we can train the model in an end-to-end way. We use the sum of classification, regression and self-supervised losses to train the model. Specifically, we use:

$$\mathcal{L} = \mathcal{L}_{cls} + \mathcal{L}_{reg} + \mathcal{L}_{terminal} + \mathcal{L}_{ss} \tag{10}$$

For classification and regression loss design, we adopt the formulation proposed in [2]. $\mathcal{L}_{terminal} = \frac{1}{N}\sum_{i=1}^{N} L2\Big((x_{i,pred}^T, y_{i,pred}^T), (x_{i,GT}^T, y_{i,GT}^T)\Big)$ is a simple L2 loss that minimizes the distance between predicted final-endpoints and the ground-truth. This is because $\mathcal{L}_{reg}$ is averaged across all time-points $1 : T$, and from a practical end user perspective, minimizing the endpoint loss is much more important than weighting loss from all time-steps equally. Our proposed pretext tasks contribute to $\mathcal{L}_{ss}$. During evaluation, we study each pretext task separately, and their corresponding loss formulations defined in Eq. (6), Eq. (7), Eq. (8), Eq. (9) are used for joint training.

## 5 Experiments

**Dataset:** Argoverse provides a large-scale dataset, where the task is to forecast 3 seconds of future motions, given 2 seconds of past observations. It has more than 300K real-world driving sequences collected in Miami (MIA) and Pittsburgh (PIT). Those sequences are further split into train, validation, and test sets, without any geographical overlap. Each of them has 205,942, 39,472, and 78,143 sequences respectively. In particular, each sequence contains the positions of all actors in a scene within the past 2 seconds history, annotated at 10Hz. It also specifies one actor of interest in the scene, with type 'agent', whose future 3 seconds of motion are used for the evaluation. The train and validation splits additionally provide future locations of all actors within 3 second horizon labeled at 10Hz, while annotations for test sequences are withheld from the public and used for the leaderboard evaluation. HD map information is available for all sequences.

**Experimental Details:** To normalize the data, we translate and rotate the coordinate system of each sequence so that the origin is at current position $t = 0$ of 'agent' actor and x-axis is aligned with its current direction, i.e., orientation from the agent location at $t = -1$ to the agent location at $t = 0$ is the positive x axis. We use all actors and lanes whose distance from the agent is smaller than 100 meters as the input. We train the model on 4 TITAN-X GPUs using a batch size of 128 with the Adam [43] optimizer with an initial learning rate of $1 \times 10^{-3}$, which is decayed to $1 \times 10^{-4}$ at 100,000 steps. The training process finishes at 128,000 steps and takes about 10 hours to complete. We provide more implementation details in the supplementary.

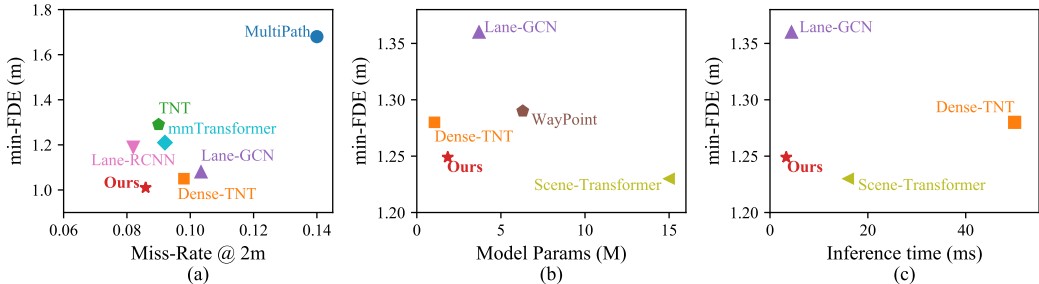

Figure 3: (a) min-FDE$_6$ - Miss-Rate$_6$ trade-off on Argoverse Validation. Lower-left is better. We optimize both successfully in comparison to other popular approaches. (b) and (c) We plot min-FDE on Argoverse Test against number of model parameters (in millions) and inference time (in milliseconds). We find that there is a trade-off between min-FDE performance, architectural complexity (as measured by number of parameters) and computational efficiency (as measured by inference time). Our work achieves the best trade-off (lower-left).

## 6 Results

### 6.1 Ablation Studies

We first examine the effect of incorporating our proposed pretext tasks (Sec. 4) with the standard data-driven motion forecasting baseline (Sec. 3). While evaluating the importance of our proposed pretext tasks, we wish to underline that motion prediction for autonomous driving is a safety-critical task, especially at intersections where most of our data is collected, and most accidents also happen. We thus posit that in this situation, even a small error in predicting final locations (FDE) for a given agent can lead to dangerous potential collision scenarios. Results in Tab. 2 show that all proposed pretext tasks improve motion forecasting performance for Argoverse. Specifically, the Lane Mask pretext task improves min-FDE by 8.9% and MR@2m by 20.3%. Distance to Intersection improves min-FDE by 7.1% and 19.3%. Maneuver classification improves min-FDE by 6.3% and MR@2m by 15.4%. We expect that improving the quality of clustering for maneuvers and thus creating better pseudo-labels will improve this further. Finally, Success/Failure classification improves min-FDE by 9.8% and, perhaps expectedly, MR@2m by 22.4%. Moreover, since pretext tasks are not used for inference and only for training, they also do not add any extra parameters or FLOPs to the baseline, thereby increasing accuracy but at no cost to computational efficiency or architectural complexity.

### 6.2 Comparison with State-of-the-Art

**Performance:** We compare our approach with top entries on Argoverse [1] in Tab. 3. SSL-Lanes improves the metrics for $K = 1$ convincingly and outperforms existing approaches w.r.t. *min-ADE$_1$*, *min-FDE$_1$* and *MR$_1$*. We are strongly competitive w.r.t. *min-ADE$_6$*, *min-FDE$_6$* and *MR$_6$*. with a relatively simple architecture.

**Trade-off between min-FDE and Miss-Rate:** *min-FDE$_6$* and *MR$_6$* are both important for autonomous robots to optimize. Ideally we wish for both of these metrics to be low. However, there exists a frequent trade-off between them. We compare this trade-off in Fig. 3(a) w.r.t 6 other popular motion forecasting models (in terms of citations and GitHub stars), namely: Lane-GCN [2], Lane-RCNN [2], MultiPath [3], mm-Transformer [9], TNT [5] and Dense-TNT [6] on the Argoverse Validation Set. We are on the lowest-left of Fig. 3(a), meaning we optimize both *min-FDE$_6$* and *MR$_6$* successfully in comparison to other top models.

**Trade-off between accuracy, efficiency and complexity:** We are the first to point out a trade-off that exists for current state-of-the-art motion forecasting models between forecasting performance, architectural complexity and inference speed, in this work. This is illustrated in Fig. 3(b, c). In contrast to the popular models, our approach has high accuracy (*min-FDE$_6$*: 1.25m, *MR$_6$*: 13.3%), while also having low architectural complexity (1.84M parameters) and high inference speed (3.3 ms). Thus it provides a great balance for application to real-time safety-critical autonomous robots.

| Description | Experimental Setup | | Method | $minADE_6$ | $minFDE_6$ | $MR_6$ |
| | Training | Validation | | | | |
|---|---|---|---|---|---|---|
| Effects of limited training data | 25% of train | All | Baseline | 0.82 | 1.33 | 14.66 |
| | | | Ours | **0.78** | **1.22** | **12.63** |
| Effects of new domain | 100% PIT + 20% MIA | MIA val | Baseline | 0.88 | 1.46 | 17.21 |
| | | | Ours | **0.85** | **1.34** | **14.96** |
| Performance on difficult maneuvers | All | Turning & lane changing | Baseline | 0.90 | 1.53 | 19.90 |
| | | | Ours | **0.84** | **1.34** | **14.93** |
| Effects of imbalanced data | 2x straight 1x other maneuvers | Turning & lane changing | Baseline | 0.94 | 1.65 | 21.53 |
| | | | Ours | **0.90** | **1.49** | **17.97** |
| Effects of noisy data | All | Gaussian noise ($\sigma = 0.2$) with $p = 0.25$ | Baseline | 1.01 | 1.37 | 15.59 |
| | | | Ours | **0.96** | **1.24** | **11.98** |
| Effects of noisy data | All | Gaussian noise ($\sigma = 0.2$) with $p = 0.5$ | Baseline | 1.19 | 1.56 | 20.64 |
| | | | Ours | **1.13** | **1.40** | **15.65** |

Table 4: Different experimental settings for SSL-based training

### 6.3 When does SSL help Motion Forecasting?

We design 6 different training and testing setups as shown in Tab. 4. We use Success/Failure classification as the pretext task, and all models are trained for 50,000 steps. We initialize the map-encoder with the parameters from a model trained with the lane-masking pretext task.

We hypothesize that training with SSL pretext tasks helps motion forecasting in the following ways: (a) Topology-based context prediction assumes feature similarity or smoothness in small neighborhoods of maps, and the resulting feature representation may improve prediction performance. This is mainly expected to help in the first and second settings, which requires generalizing to new topologies. (b) Clustering and classification assumes that feature similarity implies target-label similarity and can group distant nodes with similar features together, leading to better generalization. This is mainly expected to help with dataset imbalance and performance on difficult maneuvers, which requires generalizing to hard cases. (c) Supervised learning with imbalanced datasets sees significant degradation in performance. Although most of the data samples in Argoverse are at an intersection, a significantly large number involve driving straight while maintaining speed. Recent studies [44] have shown that SSL tends to learn richer features from more frequent classes, which also allows it to generalize to other classes better. We expect this to help with imbalanced data, limited training data and noisy data.

**SSL leads to better generalization compared to pure supervised learning:** To provide evidence for our hypotheses, we design 6 different training and testing setups as shown in Tab. 4. We use Success/Failure classification as the pretext task, and all models are trained for 500,000 steps. We initialize the map-encoder with the parameters from a model trained with the lane mask pretext task. There is strong evidence from our experiments that SSL-based tasks provide better generalization and can thus be more effective than pure supervised training.

## 7 Conclusion

We propose SSL-Lanes to leverage supervisory signals generated from data for free in the form of pseudo-labels and integrate it with a standard motion forecasting model. We validate our proposed approach by achieving competitive results on the challenging large-scale Argoverse benchmark. We further demonstrate that each proposed SSL pretext task improves upon the baseline, especially in difficult cases like left/right turns and acceleration/deceleration. We also provide hypotheses and experiments on why SSL-Lanes can improve motion forecasting.

**Limitations:** The different losses for are only used in a 1:1 ratio without tuning them. We also use only one pretext task at a time and do not explore the combination of these different tasks. For our future work, we plan to incorporate meta-learning [45] to identify an effective combination of pretext tasks and automatically balance them. Another limitation is that we report improvements with SSL-pretext tasks in scenarios without specifically considering multiple heavily interacting agents. In the future we would like to explore how the interactions between road agents can influence our SSL losses on the interaction split of the Waymo Open Motion dataset (WOMD) [46]. We would also like to study the generalization of our work to other datasets without re-training.

## Acknowledgments

This research was funded by the Mitacs Accelerate Program and Gatik Inc. This article solely reflects the opinions and conclusions of its authors.

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
