# OpenReview forum: "SSL-Lanes: Self-Supervised Learning for Motion Forecasting in Autonomous Driving"
_robot-learning.org/CoRL/2022/Conference — CoRL 2022 Poster_

### Official Review · Reviewer_QGzU · 2022-07-18

**Originality:** Good
**Technical Quality:** Good
**Clarity Of Presentation:** Good
**Impact:** 3

**Recommendation:**

Weak Accept: I recommend accepting the paper, but will not argue for my recommendation if the majority of other reviewers have a different opinion.

**Summary:**

This paper considers motion forecasting for autonomous driving (i.e., predicting the future states of road users). The paper proposes four novel loss terms to be used in combination with a traditional loss (e.g., mean squared error of future agent positions). The four losses are: "lane masking" to learn lane structure, "distance to intersection", "maneuver classification", and "success/failure" to predict trajectory endpoints. Where traditional losses are supervised from the labeled/ground-truth data, the proposed losses are self-supervised in the sense that they leverage pseudolabels created by additional processing of the original data. The method assumes access to ground truth data of road user positions and road structure. Training with these losses increases accuracy in future motion forecasts on the Argoverse dataset; the utility of each individual loss is confirmed by an ablation study. Overall, these losses improve performance over the state of the art.

**Issues:**

Please see the "weaknesses" section for major issues. Minor issues as follows:

104: "BEV" (presumably bird's eye view) should be spelled out for unfamiliar readers

The lane masking and distance-to-intersection tasks use human-supplied labels of lane structure, so are they really self-supervised? One could certainly argue that labeling lanes is less work or easier than labeling individual objects in video frames, for example. Note, the maneuver classification and success/failure classification tasks do generate pseudolabels, so I would consider them both to be self-supervised.

**Quality Of The Limitations Section:**

Limitations section not present

**Reviewer Expertise:**

4: The reviewer is confident but not absolutely certain that the evaluation is correct

**Robotics Focus:**

Highly relevant to robotics but no hardware experiments

**Strengths And Weaknesses:**

Strengths:

- The main strength of this paper is in improving performance by crafting novel losses out of existing data. This is important for two reasons: (1) no additional hand-labeling is required, and (2) inference time does not change because these losses could be used with any model architecture.

- Another key strength of this paper is that the text is fully self-contained, clear, and well-scoped. It outlines a high-level and relevant challenge in robot autonomy, proposes a method to address the challenge, and thoroughly discusses why the proposed method worked.

- A third strength of this paper is in its thorough ablation study. The paper clearly summarizes exactly what performance increase each novel loss provides. Furthermore, the paper presents some well-reasoned hypotheses (and more detailed supplementary discussion) about why these losses increase performance.

- Finally, from a quick look through the literature, I agree that there are no other works that thoroughly investigate self-supervision in motion forecasting. I found the following papers [R1,R2,R3].


Weaknesses:

- One weakness (potentially major) of this paper is that the results are limited to a single dataset. While the Argoverse dataset is "large scale," it would be helpful to add some discussion of why this one dataset is sufficient, or why other datasets may not be feasible to use (e.g., NuScenes or Waymo).

- A second weakness (minor) of this paper is that it ignores the case of motion forecasting for interaction simulation. Motion forecasting models are most useful when they can act as simulators for realistic interaction scenarios, which really requires them to generalize (which is, I realize, always a complaint with any neural network model). My understanding of the literature is that models, especially when trained on lane information and road structure, often memorize what they have seen without learning a generic model of how agents actually move around. The current losses explicitly encourage a motion forecasting model to stay near ground truth and potentially memorize the data, so could they inhibit the model's ability to generalize?

- A third (very minor) weakness of the paper is that its self-supervised losses still rely on extensive existing labeled data.

- Finally (major), there is no easy-to-find limitations section within the main document, though some discussion of limitations is spread throughout the results section. The supplementary material also provides a very brief discussion. It would be great if the paper could clearly discuss its limitations, and ideally refute the weaknesses I note above, within the main text.

Update: The paper has been revised to clearly address these concerns, so I am happy to upgrade to my recommendation.


References:

[R1] Janjoš, F., Dolgov, M. and Zöllner, J.M., 2021, July. Self-Supervised Action-Space Prediction for Automated Driving. In 2021 IEEE Intelligent Vehicles Symposium (IV) (pp. 200-207). IEEE.

[R2] Hu, P., Huang, A., Dolan, J., Held, D. and Ramanan, D., 2021. Safe local motion planning with self-supervised freespace forecasting. In Proceedings of the IEEE/CVF Conference on Computer Vision and Pattern Recognition (pp. 12732-12741).

[R3] Luo, C., Yang, X. and Yuille, A., 2021. Self-supervised pillar motion learning for autonomous driving. In Proceedings of the IEEE/CVF Conference on Computer Vision and Pattern Recognition (pp. 3183-3192).

**Summary Of Recommendation:**

This paper basically proposes some task-specific loss functions to improve performance on a benchmark in simulation; in that regard, though the results are nice and the text is well written, the underlying story is not very exciting.

I believe the paper can be greatly improved (i.e., switch from reject to accept) by including a fuller discussion of the importance of this work in a broader context. What are the limitations? Can the proposed method be used for high-quality interaction simulation? Can the proposed method generalize to other datasets, or be demonstrated in real-world use?

Update: The authors addressed my concerns well. I think the paper is not revolutionary, but definitely useful and important, hence my "weak accept."

---

> ### Author Response · Authors · 2022-08-27
> **Author response to Reviewer QGzU (Part 1/2)**
>
> Thank you for your detailed and insightful comments! We are pleased to hear that you found our study to be thorough, clear, and well-scoped. You brought up great questions and suggestions, which we discuss further below. We have also made some revisions to the paper (highlighted in red).
>
> Please note that our response has been split into two parts due to space constraints. (Part 1/2)
>
> $~$
>
> **Q1:** "One weakness (potentially major) of this paper is that the results are limited to a single dataset. While the Argoverse dataset is "large scale," it would be helpful to add some discussion of why this one dataset is sufficient, or why other datasets may not be feasible to use (e.g., NuScenes or Waymo)."
>
> **A1:** This is a good point. In our study, which focuses on the effect of SSL-losses to address the shortcomings of a current state-of-the-art data-driven motion forecasting baseline, we have two main requirements for the dataset:
> 1. *Scale of Data*: Modern motion forecasting methods and self-supervised learning systems require a large amount of training data to imitate human maneuvers in complex real-world scenarios. Thus, the dataset should be large-scale and diverse, such that it has a wide range of behaviors and trajectory shapes across different geometries represented in the data.
> 2. *Interesting Scenarios for Forecasting Evaluation*: The dataset should be collected for interesting behaviours by biasing sampling towards more complex observed behaviours (e.g., lane changes, turns) and road features (e.g., intersections), since we wish to focus and improve on these cases.
>
> We now compare the commonly used motion-forecasting datasets, i.e., nuScenes, Waymo-Open-Motion-Dataset (WOMD) and Argoverse, and evaluate them based on our above-mentioned requirements. We then proceed to make a case for why Argoverse is sufficient for our study.
> * *Scale of Data:* We first compare the dataset size. We note that Argoverse is not only two orders larger than nuScenes, but also has greater number of training samples and unique trajectories compared to WOMD.
> |       | nuScenes | Waymo (WOMD) | Argoverse |
> | :---:        |    :----:   |          :---:  |  :---: |
> | Number of Unique Tracks:      | 4.3k    | 7.65m | 11.7m  |
> | Number of Training Segments:  | 1k | 104k | 324k    |
>
> * *Interesting Scenarios for Forecasting Evaluation:* We next compare if the datasets specifically mines for interesting scenarios, which is the area we want to improve the current baseline. nuScenes was not collected to capture a wide diversity of complex and interesting driving scenarios. WOMD on the other hand specifically mines for pairwise interaction scenarios, where the main objective is to improve forecasting for interacting agents. However, the scope of our study is to primarily focus on motion at intersections undergoing lane-changes and turns. We expect the SSL-losses to improve understanding of the context/environment, trajectory embeddings and address data-imbalance with respect to maneuvers. We leave heavy interaction-based use cases for future work. Finally, Argoverse mines for interesting motion patterns at intersections, which involve lane-changes, acceleration/deceleration, and turns. We thus find this dataset best suited to showcase the benefits of our proposed method.
>
> * *Community focus on Argoverse:* We also find that many popular motion forecasting methods published by the robotics community have also included evaluations only on the Argoverse dataset including: [Lane-GCN](https://arxiv.org/pdf/2007.13732.pdf),  [Lane-RCNN](https://arxiv.org/abs/2101.06653), [PRIME](https://arxiv.org/abs/2103.04027), [DCMS](https://arxiv.org/abs/2204.05859), [TPCN](https://arxiv.org/abs/2103.03067),  [mm-Transformer](https://arxiv.org/pdf/2103.11624.pdf), [HiVT](https://openaccess.thecvf.com/content/CVPR2022/papers/Zhou_HiVT_Hierarchical_Vector_Transformer_for_Multi-Agent_Motion_Prediction_CVPR_2022_paper.pdf), [Multi-modal Transformer](https://arxiv.org/abs/2109.06446), [DSP](https://arxiv.org/abs/2111.01592) etc. This makes it easier for us to position our work with respect to these approaches.
>
> We have also included this discussion in our *Experiments* and *Limitations* section of the paper.
>
> $~$
>
> **Q2:** Finally (major), there is no easy-to-find limitations section within the main document, though some discussion of limitations is spread throughout the results section. The supplementary material also provides a very brief discussion. It would be great if the paper could clearly discuss its limitations, and ideally refute the weaknesses I note above, within the main text.
>
> **A2:** We have updated the paper to include an explicit *Limitations* section where we clearly discuss the shortcomings of our method.

---

> > ### Author Response · Authors · 2022-08-27
> > **Author response to Reviewer QGzU (Part 2/2)**
> >
> > Please note that our response has been split into two parts due to space constraints. (Part 2/2)
> >
> > $~$
> >
> > **Q3:** My understanding of the literature is that models, especially when trained on lane information and road structure, often memorize what they have seen without learning a generic model of how agents actually move around. The current losses explicitly encourage a motion forecasting model to stay near ground truth and potentially memorize the data, so could they inhibit the model's ability to generalize?
> >
> > **A3:** We believe that the following construction enables a generic understanding of how an object moves in a given environment without memorizing the training data:
> > 1. Each area of a scene corresponds to a specific numeric range of input coordinates, which may lead to a neural network associating these areas with certain motion patterns. To prevent this, we centre around the agent of interest and normalize all other trajectory and map coordinates with respect to it.
> > 2. We predict relative motion as opposed to absolute motion for the future trajectory. This helps to learn general motion patterns.
> > 3. Reconstructing the map or predicting distances from map elements are conducted in a frame-of-reference relative to the agent of interest. This helps in learning general map connectivity.
> > 4. Following work in pedestrian trajectory prediction, we also additionally add random rotations to the training trajectories to reduce directional bias.
> >
> > Thus we believe the proposed method can be used for simulating realistic motion in other environments.
> >
> > $~$
> >
> > **Q4:**  (a) Self-supervised losses still rely on extensive existing labeled data.
> > (b) The lane masking and distance-to-intersection tasks use human-supplied labels of lane structure, so are they really self-supervised? One could certainly argue that labeling lanes is less work or easier than labeling individual objects in video frames, for example. Note, the maneuver classification and success/failure classification tasks do generate pseudo-labels, so I would consider them both to be self-supervised.
> >
> > **A4:** We admit that our crafted SSL-losses rely on labeled information like location of lane centres, intersections and agent end-points. However we find that most modern motion forecasting benchmarks like NuScenes, Argoverse, WOMD etc., provide richly annotated semantic information with the datasets. In this work thus, we decide to use the readily available information to improve upon the shortcomings of the state-of-the-art forecasting model. We will leave the application of using unannotated information (for e.g., inferring lane-boundaries and intersections from visual map information) for future work.

---

> > > ### Author Response · Authors · 2022-08-27
> > > **Paper Revision in response to Reviewer QGzU**
> > >
> > > **Comment:**
> > >
> > > We have attached the revised paper and supplementary material in the PDFs below.
> > >
> > > **Zip File:**
> > >
> > > /attachment/9d75708d545881024f02e5d660a826b28f681990.zip

---

### Official Review · Reviewer_fptR · 2022-07-31

**Originality:** Good
**Technical Quality:** Good
**Clarity Of Presentation:** Very Good
**Impact:** 3

**Recommendation:**

Weak Accept: I recommend accepting the paper, but will not argue for my recommendation if the majority of other reviewers have a different opinion.

**Summary:**

This work leverages the potential of self-supervised learning (SSL) to boost representation learning for trajectory prediction. Four pretext tasks are proposed for joint training in trajectory prediction, including lane masking, distance to intersection learning, maneuver classification, and target hitting classification. By incorporating these SSL tasks with typical prediction frameworks, the prediction performance gains without increasing the model complexity and parameter number. The target-hitting classification shows the most prominent performance improvement among all the pretext tasks.

**Issues:**

Some minor issues are listed below:
- As I know CS-LSTM had used a similar set of maneuver classification for trajectory prediction in highway scenarios, but in the urban scenarios of Argoverse, is that really enough to cover the driving maneuvers in urban driving (overtaking, cut-in, nudging)? Have you tried more clusters and what about the results?
-  It is suggested to draw Fig3.a with the performance on the Argoverse test set, which would reflect more difference. It's better to supplement more model results on Figure3.b and Figure3.c (for example, TNT results should be added) to make your conclusion convincing.

**Quality Of The Limitations Section:**

Limitations are addressed clearly

**Reviewer Expertise:**

5: The reviewer is absolutely certain that the evaluation is correct and very familiar with the relevant literature

**Robotics Focus:**

Highly relevant to robotics but no hardware experiments

**Strengths And Weaknesses:**

Strengths:
- This work exploits the SSL power to enhance the capability of representation learning from different aspects, which may inspire many more approaches for optimizing trajectory prediction frameworks in this direction.
- The main structure is built upon Land-GCN. All of the pretext tasks bring some performance improvement, in which the success/failure classification gains most with a simple design.

Weaknesses:
- The reviewer admits that this is the first work to study the SSL tasks in trajectory prediction systematically, but actually, some of the pretext tasks had been proposed in prior works without being termed as SSL. This is the primary weakness of this paper. Specifically, the lane masking task is used as an auxiliary task in VectorNet[12]. The maneuver classification together with "classification+regression" is proposed in CS-LSTM, Multipath[3], etc., to make multimodal predictions. Considering the existing similar tries and the fact that all these SSL tasks contribute to the prediction performance, it is better to make an effective combination of them so that this work would be more solid.
- The reviewer believes that the tradeoff between performance and model complexity seems to be overclaimed. Boosting the model performance without increasing model complexity and parameter number should be the common advantage of self-supervised learning, but not from any specific design from this work. Secondly, the reviewer wonder if "the best accuracy-simplicity-efficiency" is overstated, as the critical ranking metrics in Argoverse are not the best while some methods are absent in the comparison of Model Params and Inference time.



**Summary Of Recommendation:**

Considering the fact that 1) the Lane-masking task and the Maneuver classification task had been proposed in previous works, and 2) the best trade-off between complexity and efficiency are drawn from part of the methods, the reviewer believes this work still needs to improve. An ideal direction to make it solid and convincing is to combine different pretext tasks effectively.

---

> ### Author Response · Authors · 2022-08-27
> **Author response to Reviewer fptR (Part 1/2)**
>
> Thank you for your valuable feedback! We really appreciate that you found our work could inspire many more approaches for optimizing trajectory prediction frameworks, and your remark that this is the first work to study the SSL tasks in trajectory prediction systematically. We respond to your main concerns individually below, and have made revisions (highlighted in blue) based on your comments, which we believe has improved our work.
>
> Please note that our response has been split into two parts due to space constraints. (Part 1/2)
> $ $
>
> **Q1:** Actually, some of the pretext tasks had been proposed in prior works without being termed as SSL. This is the primary weakness of this paper. Specifically, the lane masking task is used as an auxiliary task in VectorNet[12]. The maneuver classification together with "classification+regression" is proposed in CS-LSTM, Multipath[3], etc., to make multimodal predictions.
>
> **A1:** We think this is an important point and welcome this opportunity to distinguish our work from the methods mentioned in your review.
> 1. SSL-Lanes vs. VectorNet [12]: Vector-Net is the only other motion forecasting work that proposes to randomly mask out the input node features belonging to either scene context or agent trajectories, and ask the model to reconstruct the masked features. Their intuition is to encourage the graph networks to better capture the interactions between agent dynamics and scene context. However, our motivation differs from VectorNet in two respects: (a) We propose to use masking to learn local map-structure better, as opposed to learning interactions between map and the agent. This is an easier optimization task, and we out-perform VectorNet. (b) A lane is made up of several nodes. We propose to randomly mask out a certain percentage of each lane. This is a much stronger prior as compared to randomly masking out any node (which may correspond to either a moving agent or map) and ensures that the model pays attention to all parts of the map.
>
> 2. SSL-Lanes vs. CS-LSTM [20]: CS-LSTM appends the encoder context vector with a one-hot vector corresponding to the lateral maneuver class and a one-hot vector corresponding to the longitudinal maneuver class. Subsequently, the added maneuver context allows the decoder LSTM to generate maneuver specific probability distributions. This construction however is quite different from our work because it is not auxiliary in nature - it always outputs and appends a maneuver to the decoder, even during inference. This we believe is too strong of a bias for the prediction model, especially given the fact that the maneuvers are generated using very simple velocity profiles and not from careful mining of the data. In our conditioning, the maneuvers are mined from data and the final motion prediction does not depend directly on them. We believe this design is much more flexible since it allows to generate more supervisory signals in the form of maneuvers during training, but at the same time does not require an explicit maneuver to condition the final future forecast trajectory output during inference.
>
> 3. SSL-Lanes vs. MultiPath [3]: MultiPath is also not auxiliary in nature: it factorizes motion uncertainty into intent uncertainty and control uncertainty; models the uncertainty over a discrete set of intents with a softmax distribution; and then outputs control uncertainty as a Gaussian distribution dependent on each waypoint state of the anchor trajectory (corresponding to the intent). While this construction is highly intuitive and effective by design, it is very different from our SSL-based construction. Ours is an auxiliary task which provides supervision during training, and effectively functions as a regularizer, while being general enough to be used with any other data-driven motion forecasting model.
>
> While we agree that the intuitions of the above-mentioned three methods are probably similar to ours, we believe our work is sufficiently novel in terms of motivation, construction, implementation, as well the results presented.
>
>
> $ $
> **Q2:** Considering the existing similar tries and the fact that all these SSL tasks contribute to the prediction performance, it is better to make an effective combination of them so that this work would be more solid.
>
> **A2:** We agree that an effective combination would make the work more solid, and mention this in our *Limitations and Future Work* section. This combination of SSL-tasks is however not trivial by design. For example, an interesting direction would be to learn weight functions to softly select auxiliary tasks and balance them with the primary forecasting task via meta-learning inspired by [45]. Regrettably, we were not able to add an experiment like this due to time constraints.

---

> > ### Author Response · Authors · 2022-08-27
> > **Author response to Reviewer fptR (Part 2/2)**
> >
> > Please note that our response has been split into two parts due to space constraints. (Part 2/2)
> >
> > $ $
> >
> > **Q3:** Boosting the model performance without increasing model complexity and parameter number should be the common advantage of self-supervised learning, but not from any specific design from this work.
> >
> > **A3:** While the novelty of our work lies in scoping out loss functions which can use pseudo-labels and improve upon the state-of-the-art on challenging real-world forecasting samples, we feel that boosting the model performance without increasing model complexity and parameter number has not been the focus of discussion in most of the papers we reviewed in the literature. We thus feel that our discussion would add value to this community.
> >
> > $ $
> >
> > **Q4:** The reviewer wonder if "the best accuracy-simplicity-efficiency" is overstated, as the critical ranking metrics in Argoverse are not the best while some methods are absent in the comparison of Model Params and Inference time.
> >
> > **A4:** We also believe that a fair comparison of the performance between neural network methods should ensure that each architecture is parameter-matched and the training is FLOP-matched. However, the
> > Argoverse leaderboard does not require entrants to report their parameter or FLOP counts, which is why some of the methods do not have this information available publicly. We operate on the model information that was available to us at time of submission, and highlight this as an interesting outcome of our proposed method.

---

> > > ### Author Response · Authors · 2022-08-27
> > > **Paper Revision in response to Reviewer fptR**
> > >
> > > **Comment:**
> > >
> > > We have attached the revised supplementary material in the PDFs below.
> > >
> > > **Zip File:**
> > >
> > > /attachment/f648aa63d83cfb806da5d1db0c82a3747d4e6632.zip

---

### Official Review · Reviewer_dvVA · 2022-07-31

**Originality:** Good
**Technical Quality:** Very Good
**Clarity Of Presentation:** Excellent
**Impact:** 3

**Recommendation:**

Weak Accept: I recommend accepting the paper, but will not argue for my recommendation if the majority of other reviewers have a different opinion.

**Summary:**

This paper proposes to use self-supervised learning to improve trajectory prediction models. More specifically, it proposes four novel self-supervised tasks for motion forecasting.

**Issues:**

No major issues. Some minor improvements are described above.

**Quality Of The Limitations Section:**

Limitations are not well addressed

**Reviewer Expertise:**

4: The reviewer is confident but not absolutely certain that the evaluation is correct

**Robotics Focus:**

Highly relevant to robotics but no hardware experiments

**Strengths And Weaknesses:**

Strengths:
To my knowledge, this is the first approach to exploit self-supervised learning in motion prediction tasks. The presented results clearly show that self-supervised learning enables increasing the motion forecasting models' performance without a significant increase in the model's complexity and inference times. Overall, the paper is well written, it presents a good review of the SoA and strong experimental results.

Weaknesses:
There are some minor improvements to improve this paper's readability.
1 - In Fig.1., The baseline [2] does not seem to miss all the predictions. The second case seems to be a good prediction. I would recommend the authors revise this statement. Additionally, the figures are ver
2 - Also in Fig.1., please discuss the result for Task 4, the second sub-figure counting from the top. It seems that the proposed method missed predicting a correct mode close to the GT. Why is this the case? Please discuss.
3 - In Fig.2, the caption does not fully describe the figure. In my opinion, a figure should be self-explainable. I recommend the authors to explain the figure in the caption.
4 - Adding an accompanying video to present the qualitative results would increase this paper's contribution.
5 - In Eq. 2 and Eq.1. not all the variables are presented/introduced. Please make sure that all the variables are explained. The same for Eq. 4, what is L_sup?
6 - In Sec. 4.1.3., which maneuvers are considered? How is the output classification vector defined?
7 - In Sec. 6.3., line 251, the presented results do not support the statement "allows it to generalize to other classes better". Adding evaluation results of the proposed method on new datasets that were not used for training to evaluate how self-supervised learning enables improving generalization would be a significant result. The authors may consider to add such evaluation in the paper or appendix section.

**Summary Of Recommendation:**

To my knowledge, this is the first approach to exploit self-supervised learning in motion prediction tasks. Although the contribution is incremental, I think the presented results are significant and send a clear message to the community that self-supervised learning can also be employed to improve motion forecasting models. Therefore, I would recommend to accept this paper.

---

> ### Author Response · Authors · 2022-08-28
> **Author response to Reviewer dvVA**
>
> Thank you for your detailed and valuable feedback! We really appreciate that you found our work to be well-written and novel. We respond to your main comments individually below and have made revisions to the paper based on your comments, which we believe has improved the paper.
>
> **Q1:** The baseline [2] does not seem to miss all the predictions. The second case seems to be a good prediction. I would recommend the authors revise this statement.
>
> **A1:** We make the statement that the baseline misses the prediction because the nearest prediction is at-least 9m away from the ground-truth, whereas according to motion prediction evaluation metrics, it is considered to be missed if it is more than 2m away.
>
> **Q2:** Also in Fig.1., please discuss the result for Task 4, the second sub-figure counting from the top. It seems that the proposed method missed predicting a correct mode close to the GT. Why is this the case? Please discuss.
>
> **A2:** This is an interesting observation. This sample corresponds to accelerating at a turn, but also has a noisy past history. We believe that the pretext task 4 imposes stronger regularization, thereby causing it to predict modes which are more represented in the data for this past history and corresponding geometry. We add a detailed description of Fig 1 and the issue that you mentioned in the supplementary.
>
> **Q3:** In Fig.2, the caption does not fully describe the figure. In my opinion, a figure should be self-explainable. I recommend the authors to explain the figure in the caption
>
> **A3:** We agree, and add a more detailed caption for this figure.
>
> **Q4:** Please make sure that all the variables are explained. The same for Eq. 4, what is L_sup?
>
> **A4:** We defined the variables but because of space constraints, some details have been moved to the supplementary section. For example, L_sup has been defined in Line-109 and denotes the supervised loss calculated between the predicted output and the ground-truth future trajectories.
>
> **Q5:** In Sec. 4.1.3., which maneuvers are considered? How is the output classification vector defined?
>
> **A5:** We use the following maneuvers: {maintain-speed, accelerate, decelerate, turn-left, turn-right, lane-change}. We provide details about how we define and mine the intention/the output classification vector (one-hot encoding of intentions) in Section 7.3 in the supplementary. For this pretext task, we first divide the lateral and longitudinal maneuvers by choosing a threshold angle of 20 degrees from the vertical. We next find that constrained k-means on agent end-points. for lateral and longitudinal maneuvers works best to separate the trajectory samples into different clusters. For differentiating the longitudinal maneuvers from the lane-change maneuver, we check a combination of the distance from the lane centerlines for start and stop positions and the orientations of the nearest centerline for start and stop positions.
>
> **Q6:** In Sec. 6.3., line 251, the presented results do not support the statement "allows it to generalize to other classes better". Adding evaluation results of the proposed method on new datasets that were not used for training to evaluate how self-supervised learning enables improving generalization would be a significant result. The authors may consider to add such evaluation in the paper or appendix section.
>
> **A6:** We just want to clarify that we make the statement "allows it to generalize to other classes better" in reference to the recent study on SSL and data-imbalance [44]. Specifically, we say: "Recent studies [44] have shown that SSL tends to learn richer features from more frequent classes, which also allows it to generalize to other classes better. We expect this to help with imbalanced data, limited training data and noisy data. There is evidence from our experiments (Table 4) that SSL-based tasks provide better generalization compared to pure supervised training."
> We agree that the generalization experiments to a completely different dataset will be a strong result. However, we believe that this is a study in itself, which requires carefully defining which datasets we wish to generalize to. We leave this as future work.

---

### Official Review · Reviewer_b8zt · 2022-08-04

**Originality:** Very Good
**Technical Quality:** Very Good
**Clarity Of Presentation:** Very Good
**Impact:** 4

**Recommendation:**

Strong Accept: I recommend accepting the paper and will argue for my recommendation even if other reviewers hold a different opinion.

**Summary:**

The authors develop 4 self-supervision schemes for motion forecasting for autonomous driving. These are: lane masking, distance to intersection, maneuver classification, and success/failure classification. They show that, in general, these pretext tasks improve performance and allow them to achieve a good balance between performance and architectural complexity / inference time.

**Issues:**

Only small formating and presentations issues listed above.

**Quality Of The Limitations Section:**

Limitations section not present

**Reviewer Expertise:**

3: The reviewer is fairly confident that the evaluation is correct

**Robotics Focus:**

Highly relevant to robotics but no hardware experiments

**Strengths And Weaknesses:**

Strengths:

Overall I find the work very solid. SSL is a popular approach but has not been successfully applied to the important motion forecasting problem in autonomous driving. The 4 approaches are well-motivated and explained and the results are convincing.



Weaknesses / Questions:

 - I find Figure 1 very hard to parse / interpret.
 - Perhaps one point to note is that most (all?) data-driven approaches to motion forecasting are self-supervised in the sense that no hand labeling is needed. What is unique about this work is the development of the pretext tasks.

**Summary Of Recommendation:**

Overall I find the work to be a nice contribution to the area of motion forecasting that will likely have high impact.

---

> ### Author Response · Authors · 2022-08-28
> **Author response to Reviewer b8zt**
>
> Thank you for your positive and valuable feedback. We appreciate and are encouraged that you found that the work is solid.
>
> 1. Perhaps one point to note is that most (all?) data-driven approaches to motion forecasting are self-supervised in the sense that no hand labeling is needed. What is unique about this work is the development of the pretext tasks.
>
> Thanks for pointing this out! The main contribution is to identify, design and develop pretext tasks that will lead to stronger supervisory signals via pseudo-labeling.
>
> 2. I find Figure 1 very hard to parse / interpret.
>
> We were limited by space constraints and could not clearly explain which colors represented the past/ground-truth/predicted trajectories in the main-text, which is why we resorted to legends. To improve readability, we expanded on the different colors, their significance, the role of lane-centerlines etc. in detail in the Supplementary.

---

### Author Response · Authors · 2022-08-29
**We have further revised our manuscript**

**Comment:**

We have further revised our manuscript to incorporate the feedback from the detailed reviews. We make the following changes which we believe has greatly improved the work:
1. We have added a discussion for the qualitative results presented in Fig.1.
2. We have improved readability by adding a detailed caption for Fig.2.
3. We have clearly clarified the differences between our proposed work and the methods which have similar intuition but very different construction.
4. We have provided motivation for our choice of dataset used to show-case this work.
5. We have added a section that provides evidence for SSL leads to better generalization compared to pure supervised learning.
6. We have added a section that discusses the potential benefits of this work.
7. We have added an explicit limitations section which discusses short-comings and also motivates future work.

**Zip File:**

/attachment/dd7a6aad465c3fcfb5e0ed62da4a647dca4b660f.zip

---

### Meta-Review · Area_Chair_U3Py · 2022-08-14

**Recommendation:** Accept (Poster)
**Confidence:** 5

**Metareview:**

The paper studies self-supervised learning pretext tasks for motion forecasting in autonomous driving. The reviewers find the paper to be well written and address an important problem. However, the reviewers raised major concerns, most of which have been thoroughly addressed in the rebuttal. I thank the authors for the engaging discussions during the rebuttal. Some minor concerns still exist. Nevertheless, I agree with the reviewers that the paper is an interesting contribution.

**Best Paper Nomination:**

No